# Bayesian algorithm to estimate position and activity of an orphan gamma source utilizing multiple detectors in a mobile gamma spectrometry system

**Antanas Bukartas** [1] *, **Jonas Wallin** [2], **Robert Finck** [1], **Christopher Rääf** [1]

**1** Medical Radiation Physics, Lund University, Lund, Sweden, **2** Department of Statistics, Lund University, Lund, Sweden

* antanas.bukartas@med.lu.se

**Data Availability Statement:** All relevant data are within the manuscript and its Supporting information files.

## Abstract

To avoid harm to the public and the environment, lost ionizing radiation sources must be found and brought back under the regulatory control as soon as possible. Usually, mobile gamma spectrometry systems are used in such search missions. It is possible to estimate the position and activity of point gamma sources by performing Bayesian inference on the measurement data. The aim of this study was to theoretically investigate the improvements in the Bayesian estimations of the position and activity of a point gamma source due to introduction of data from multiple detectors with angular variations of efficiency. Three detector combinations were tested—a single 123% HPGe detector, single 4l NaI (Tl) detector and a 123% HPGe with 2x4l NaI (Tl) detector combination—with and without angular efficiency variations for each combination resulting in six different variants of the Bayesian algorithm. It was found that introduction of angular efficiency variations of the detectors did improve the accuracy of activity estimation slightly, while introduction of data from additional detectors lowered the signal-to-noise ratio threshold of the system significantly, increasing the stability and accuracy of the estimated source position and activity, for a given signal-to-noise ratio.

## Introduction

In situations where an ionizing radiation source is lost, the orphan source must be found as soon as possible to avoid harm to people and environment [1, 2]. Typically, mobile gamma spectrometry vehicles (cars, trucks, planes, etc.) equipped with ionizing radiation detectors are being used for such tasks [3, 4]. Most probably, such emergency missions will have limited resources (e.g. equipment, manpower, time) [5]. Usual way of performing a mobile gamma spectrometry survey of an area is travelling around the selected locality in a predetermined itinerary sampling full energy gamma counts and GPS locations at selected, regular time intervals [6]. Then the count rates in certain regions of interest (ROI) of the gamma spectra are compared to a pre-set alarm threshold level [7]. In a case where the count rate exceeds the preset alarm threshold level, an indication is made that there may be a source present nearby. To

**Funding:** This work was financially supported by the Swedish Radiation Safety Authority (SSM 2016/2274) and by the Nordic Nuclear Safety Research. Grant received by Christopher Rääf.

**Competing interests:** The authors have declared that no competing interests exist.

aid the user/operator of such systems to detect the elevated counts from the source under varying background radiation conditions, several various statistical approaches were studied [8–10]. Despite the increased probability of detection of the sources due to new background suppression methods suggested by Kock et al. or Mauring et al. [8–10], the main result of the survey remains that an increase in the count rate in a ROI in the gamma spectra indicates only that there might be a source near that measurement position [11, 12]. Such statistical approach to mobile gamma spectrometry data analysis has been utilized for decades. The thresholds can be set up by using a desired false-positive rate of detection, without sophisticated statistical knowledge or powerful computing resources, thus performing reliably in emergency situations. However, one of the problems of such a simple approach is that to further locate the source additional time and labour consuming surveys may need to be carried out—either by using a backpack or a hand held systems [13]. Sometimes, the additional survey can be severely limited by e.g. surrounding structures, foliage or even the activity of the source itself, making it very hard or almost impossible to estimate the position of the source. Also, there may be a hidden threat in the area where the missing source has been indicated. Before the area is investigated further, the support of police and radiation protection experts may be needed. It may therefore be important to know approximately where the source is. Another issue with this method is that it does not utilize all of the available data in the measurement time-series. A plot of the measurement time-series of counts in the detector will show a typical peak-like shape after the mobile system has passed a source. The shape of this peak depends only on the distance between the trajectory of the mobile gamma spectrometry vehicle and the source. The absolute height of the peak depends on the activity of the source in a given situation. Performing statistical inference on the measurement time-series could provide an estimate of the location and activity of the source based only on a number of measurements performed while driving past the source without needing to stop for additional surveys.

In a previous study [14], such a Bayesian algorithm was described and tested. The algorithm estimated the location and activity of a point gamma source by performing Bayesian inference on the data collected while driving along the road past the radioactive sources. Although it was demonstrated, that the described algorithm could potentially increase the efficiency of the current mobile gamma spectrometry methods and even reduce the doses for the first responders in emergency situations as the aforementioned additional surveys may be averted, the results also revealed some marked discrepancies between the actual and estimated positions and activities of the point gamma sources, warranting more extensive investigation of this approach. It would also be of interest to test the applicability of such Bayesian method to localise other types of sources e.g. neutron sources. As was it was observed by Nilsson et al., it would be possible to use regular gamma radiation detectors to detect the neutron radiation [15]. If appropriate physical detection model would be developed for neutron detection, it would be possible to test the applicability of Bayesian approach for neutron source search, but that is out of the scope of this paper.

The aim of this work was thus to theoretically investigate the improvements in the Bayesian estimations of the position and activity of the source due to utilization of the data from multiple detectors and implementing angular variations in counting efficiency.

## Theory

The photon fluence rate, $\dot{\varphi}$, from a gamma-emitting source at the detector can can be expressed as a function of detector-to-source distance $r$:

$$\dot{\varphi} = \frac{A n_\gamma e^{-\mu_{\mathrm{air}} r}}{4\pi r^2} \tag{1}$$

where $A$ is the activity of the source, $n_\gamma$ is the branching ratio, denoting a number of gamma photons emitted per decay, for the gamma line of interest (with primary photon energy $E_\gamma$), $\mu_{\mathrm{air}}$ is the linear air attenuation coefficient for gamma particles in air. In a case, where the detector is moving at a certain speed past the source in two dimensions (simplification of a ground vehicle moving on a flat surface) the detector-to-source vector has only two components—$r_x$, $r_y$. Substituting the two-dimensional detector-to-source vector components in the Eq 1, yields the fluence rate as a function of coordinates:

$$\dot{\varphi}(r_x, r_y) = \frac{A n_\gamma e^{-\mu_{\mathrm{air}}\sqrt{r_x^2 + r_y^2}}}{4\pi(r_x^2 + r_y^2)}.$$

(2)

For a given detector-source geometry configuration with a specific detector-to-source distance the counting efficiency of the detector, $\varepsilon$, can be expressed as a ratio between the count rate in the detector, $\dot{N}$, and the fluence rate of gamma particles at the detector, $\dot{\varphi}$. In a typical orphan source search scenario the dimensions of the active medium of the detector, $d$, is usually orders of magnitude smaller than the detector-to-source distance ($d << r$), so the incident particles can be considered as parallel. Thus, the dependence of counting efficiency on the detector-to-source distance (or more precisely—the solid angle subtended by the detector) is negligible and the counting efficiency can be considered constant throughout the detector-to-source distance, $r$. If the counting efficiency of the detector is expressed in such a way, it can be regarded as a virtual "effective detector area" and expressed as:

$$\varepsilon(E_\gamma) = \frac{\dot{N}(E_\gamma)}{\dot{\varphi}(E_\gamma)},$$

(3)

where $\dot{N}(E_\gamma)$ is the count rate in a spectral energy window (ROI) centred around $E_\gamma$, and $\dot{\varphi}(E_\gamma)$ is the fluence rate at the detector of gamma photons of energy $E_\gamma$.

Now, let's consider a point gamma source positioned at a certain distance from the road. As the mobile gamma spectrometry system is moving along the road and past a source, the detector-to-source distance and the relative angle of incidence of gamma photons will change depending on the relative position of the source and the mobile gamma spectrometry system (Fig 1). Due to additional, non uniform shielding surrounding the ionizing radiation detector (s) in real situations (e.g. a dewar for HPGe detector, photomultiplier tube with electronics or even some structural components of the vehicle), the changing angle of incidence introduces variations in the counting efficiency of the detector.

These variations can be expressed as a relative detector efficiency, $\varepsilon_{\mathrm{rel}}$, which is a dimensionless number, that multiplied by the counting efficiency for the reference direction, $\varepsilon_{\mathrm{ref}}$, gives the counting efficiency for any other selected incident direction:

$$\varepsilon(\theta, E_\gamma) = \varepsilon_{\mathrm{ref}}(E_\gamma) \cdot \varepsilon_{\mathrm{rel}}(\theta, E_\gamma).$$

(4)

It is possible to express the relative angle of incidence, $\theta$, as a difference between two-parameter arctangent functions calculated for the detector-to-source, $r$, and vehicle velocity, $v$, vector coordinates. To be able to do that, the velocity vector of the vehicle has to be evaluated using coordinates (retrieved e.g. from a GPS system) of current and previous consecutive measurements:

$$v_x = x_{\mathrm{curr}} - x_{\mathrm{prev}}; v_y = y_{\mathrm{curr}} - y_{\mathrm{prev}}.$$

(5)

Then, by subtracting the values yielded by a two-parameter arctangent function for the velocity vector from a two-parameter arctangent function for the detector-to-source vector,

## Detector-to-source angle

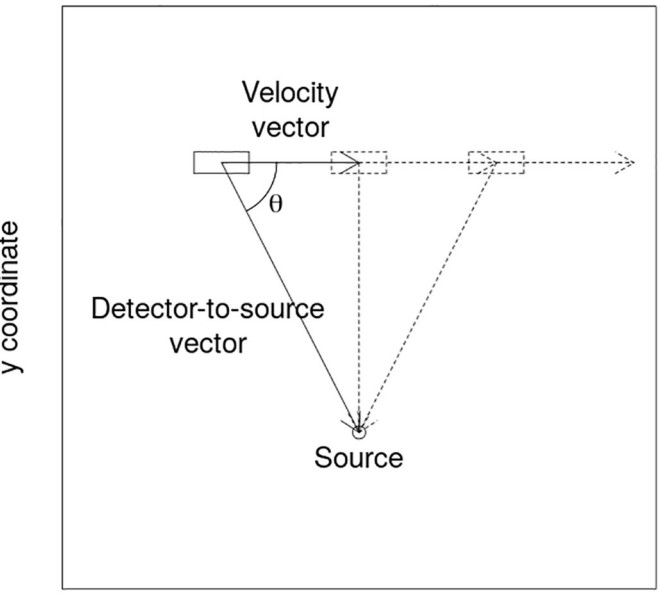

x coordinate

**Fig 1. The angle $\theta$ between the velocity vector of the mobile gamma spectrometry vehicle and the detector-to-source vector.** This angle is constantly changing when the mobile gamma spectrometry vehicle is passing the source while driving along the road.

the relative angle of incidence can be calculated:

$$\theta = \text{atan2}(v_x, v_y) - \text{atan2}(r_x, r_y), \tag{6}$$

where the $x$ and $y$ indices denote the coordinates of the respective vectors. Here, the relative angle of incidence can obtain values $\theta \in [0, 360)$ degrees (angle of 0 degrees is the driving direction). By combining the expressions for the fluence rate (Eq 2), counting efficiency (Eqs 3 and 4) and the relative angle of incidence (Eq 6), the count rate in a spectral energy window centred around energy $E_\gamma$, $\dot{N}$, can be expressed as a function of coordinates of detector-to-source and velocity vectors:

$$\dot{N}(v_x, v_y, r_x, r_y) = \frac{A n_\gamma \varepsilon_{\text{ref}} \varepsilon_{\text{rel}}(\text{atan2}(v_x, v_y) - \text{atan2}(r_x, r_y))}{4\pi} \cdot \frac{e^{-\mu_{\text{air}} \sqrt{r_x^2 + r_y^2}}}{(r_x^2 + r_y^2)}. \tag{7}$$

This equation expresses the average number of counts in a stationary detector due to a present radioactive source. If background radiation is constant in the survey area, the total count rate $\dot{N}_{\text{sum}}$ in the stationary detector, positioned some distance from the radioactive source, can be calculated as a linear combination of the average count rate due to the presence of the source $\dot{N}$ and the average background count rate in the spectral window of the primary gamma photons, $c$:

$$\dot{N}_{\text{sum}} = \dot{N} + c. \tag{8}$$

Then, let's consider a situation with a single unknown ionizing radiation source of activity $A \in \mathbb{R}^+$, with coordinates of the source $P \in S$, in an area $S \subset \mathbb{R}^2$, with a constant background

radiation giving rise to a background count rate $c$. A mobile gamma spectrometry vehicle then consecutively records number $n$ of count rates in a specific ROI in the detector $Z = (z_1, z_2, \ldots, z_n)$, measured at locations $X = (x_1, x_2, \ldots, x_n)$. If the measurements, were performed using only one detector $Z$ is a vector with elements of the vector corresponding to different measurements. If there are multiple observations using $m$ different detectors at the same locations $X$, $Z$ is then a matrix, where the element $z_{ij}$ represents the $i$-th measurement performed with $j$-th detector.

To be able to e.g. create confidence regions for the activity and position of the source a distribution of the parameters of interest given the data has to be derived. This distribution is known as the posterior distribution, which can be derived from the prior distribution (discussed in detail in [14]) and the distribution of the data given the parameters (known as the likelihood) by using Bayes theorem. The latter will be derived below.

Let $\pi(Z|X, A, P)$ denote the distribution of measurements $Z$ given the locations of the measurements $X$, the source activity $A$, and the spatial location of the source $P$, are known (the distribution of observations, given that the parameters of the source are known). In the above described case of $n$ measurements preformed with $m$ detectors, the likelihood then becomes:

$$\pi(Z|P, X, A) = \prod_{j=1}^{m} \left( \prod_{i=1}^{n} \text{Pois}(z_{ij}|\lambda_{ij} = \dot{N}_{\text{sum}_{ij}} = \dot{N}_{ij}(v_{\text{xi}}, v_{\text{yi}}, r_{\text{xi}}, r_{\text{yi}}) + c_j) \right), \qquad (9)$$

where $\lambda_{ij}$ is a mean value of a Poisson distribution of the $i$-th measurement, describing the mean number of counts in the $j$-th detector with the mean background counts $c_j$ in the $j$-th detector in that measurement, $x_i$ and $y_i$ represents $x$ and $y$ coordinates of the vectors of the $i$-th measurement. The posterior distribution can then be derived using Bayes theorem:

$$\pi(P, A|Z, X) \propto \pi(Z|P, X, A) \cdot \pi(P, A). \qquad (10)$$

where $\pi(Z|P, X, A)$ is the likelihood and $\pi(P, A)$ the prior distributions.

## Methods

An experiment studying the detection limits of various detection systems involving real $^{60}$Co (emitting two gamma photons per decay of 1.1732 and 1.3325 MeV) and $^{137}$Cs (emitting one gamma photon per decay of 0.6617 MeV) sources set up at various distances along the road was performed prior to this study (Fig 2). Sources of 5 different activities (different for each radionuclide) were used in the experiment. For a given activity of a radionuclide, the sources were set up at 4 different distances to the road. This resulted in 20 different combinations of source activities and distances—experimental setups, displayed in Table 1. A mobile gamma spectrometry vehicle (described in [14]) was then driven back and forth along the road past the sources, performing mobile gamma spectrometry measurements.

Because the scope of this article is to evaluate theoretical improvements in the Bayesian estimations, simulated data were used. To make the results comparable with an upcoming study, where an in-depth analysis of experimental results will be performed, activities of the sources, distances to the road at which the sources were positioned, GPS locations of the performed mobile gamma spectrometry measurement surveys and background radiation levels were chosen according to the real experimental conditions (Table 1) while calculating the simulated data sets.

To obtain a measure of the variation in the performance of the algorithm, it was chosen to perform Bayesian inference on 10 different realizations of counts in the detectors. Then,

## Overview of the vehicle path

**Fig 2. Overview of the vehicle path in the experiment with marked positions of the sources.** All possible locations for $^{137}$Cs and $^{60}$Co sources (marked A and B respectively) displayed with × symbol. The sources were set up at certain distances along a perpendicular line to the road.

estimated values of activity and position of the sources using six different variations of the algorithm were compared to the actual values:

- using data only from the HPGe detector with a fixed counting efficiency;

- using data only from the HPGe detector using angular variations in counting efficiency;

- using data only from the front 4l NaI(Tl) detector with a fixed counting efficiency;

- using data only from the front 4l NaI(Tl) detector using angular variations in counting efficiency;

- using data from all of the detectors present in the vehicle (123% HPGe, 2x4l NaI(Tl)) with a fixed counting efficiency for all individual detectors;

- using data from all of the detectors present in the vehicle using individual angular variations in counting efficiency for all of the detectors

All of the data referred above in description of the variations of the Bayesian algorithm is simulated data. These variations of the algorithm will be referred to as *HPGe90*, *HPGe*, *NaI90*, *NaI*, *Mult90* and *Mult* respectively further in the text. Additional description of the simulated data generation and measurement of angular variations of counting efficiency can be found in the sections below.

**Table 1. Specifications of distances from the source to the road side and activities for the experimental setups used in the field test.**

| Setup number | $^{137}$Cs Activity (MBq) | $^{137}$Cs distance (m) | $^{60}$Co Activity (MBq) | $^{60}$Co distance (m) |
|:---:|:---:|:---:|:---:|:---:|
| 1 | 47.3 | 10 | 70.6 | 20 |
| 2 | | 20 | | 30 |
| 3 | | 30 | | 50 |
| 4 | | 40 | | 70 |
| 5 | 153 | 30 | 105 | 20 |
| 6 | | 50 | | 30 |
| 7 | | 70 | | 50 |
| 8 | | 100 | | 70 |
| 9 | 518 | 50 | 302 | 50 |
| 10 | | 70 | | 70 |
| 11 | | 100 | | 100 |
| 12 | | 130 | | 130 |
| 13 | 802 | 70 | 583 | 70 |
| 14 | | 100 | | 100 |
| 15 | | 130 | | 130 |
| 16 | | 160 | | 160 |
| 17 | 1215 | 70 | 1119 | 100 |
| 18 | | 100 | | 130 |
| 19 | | 130 | | 160 |
| 20 | | 160 | | 190 |

Actual distance to the sources were 2 metres longer, due to the relative position of the road side and the detectors in the vehicle, while the vehicle is driving in a lane of the road.

## Measurement of angular variations of detector efficiency

The angular efficiency response of the mobile gamma spectrometry system [14] was measured on a patch of pavement near town of Löddeköpinge, Sweden, where a rectangular coordinate system (18 m × 11 m) was defined (Fig 3). Along the edges of the defined rectangle, 40 measurement points were chosen with distances between the points of 1.5 m (except for the last step along the shorter axis—the distance was 0.5 m), resulting in a polar resolution of roughly 10 degrees. The vehicle was then positioned in the centre of the defined coordinate system in such a way, that the centre point of the back axle of the vehicle was directly above the centre of the rectangle, with the direction of the vehicle perpendicular to the long side of the rectangle (Fig 3). In this configuration the position of all of the detectors inside the service bed of the mobile gamma spectrometry vehicle were reasonably close to the chosen centre of the coordinate system. The vehicle was in fully operational state with lid and cabin doors closed and operators present in the cabin.

The calibration was performed for the radionuclides separately—$^{60}$Co and $^{137}$Cs, using calibration sources of 34.1 ± 1.2 and 66.5 ± 2.1 MBq activity respectively. The calibration sources were positioned in the defined positions for 2-3 minutes. Then, gamma spectra in the three detectors present in the vehicle (123% HPGe, 2x4l NaI(Tl)) were measured simultaneously. Full energy peak areas in the selected ROIs (Table 2) were calculated manually. The relative displacement of the actual detector position from the defined centre of the coordinate system was measured and taken in to account when calculating the actual distance between the source and the detector.

**Fig 3. Overview of the calibration site.** The measurement vehicle is positioned inside the defined coordinate system. Picture taken from a camera drone positioned approximately above the vehicle. The coordinate system appears not parallel due to perspective. Coordinates are expressed in metres.

The counting efficiency of the detector for a given energy of gamma photons $E$ and relative angle of incidence $\theta$ was then evaluated using expression:

$$\varepsilon(E, \theta) = \frac{N4\pi r^2}{An_\gamma exp(-\mu_{air}r)t}. \tag{11}$$

where $N$ is the number of counts in the primary gamma peak, $r$ is the actual distance between the source and the detector, $A$ is the activity of the source, $n_\gamma$ is the branching ratio of the source, $\mu_{air}$ is the coefficient of gamma particles attenuation in air and $t$ is the measurement time.

Then, the resulting angular variations of efficiency for the [0;360) degree interval were interpolated each 10 degrees to obtain the angular variations of efficiency at a regular intervals of 10 degrees. Because both NaI detectors had no structural components hindering their performance at 90 degrees, the relative angular efficiencies were normalized to their 90 degree values. The same normalization could not be applied to the HPGe relative angular efficiencies—in the experiment there was a clear line of sight between the source positioned at almost 90 degrees from the HPGe detector but in the measured relative angular efficiencies there is a dip

**Table 2. Selected ROIs in the gamma spectra and their respective reference efficiencies for the $^{137}$Cs and $^{60}$Co radionuclides for different detectors used in the mobile gamma spectrometry vehicle.**

| Detector | $^{137}$Cs | | $^{60}$Co | |
|---|---|---|---|---|
| | ROI (keV) | Ref. Eff. (m$^2$) | ROI (keV) | Ref. Eff. (m$^2$) |
| 123% HPGe | 658-665 | 0.0021 | 1327-1337 | 0.0015 |
| Front NaI(Tl) | 600-750 | 0.0254 | 1247-1470 | 0.0166 |
| Rear NaI(Tl) | 600-750 | 0.0261 | 1247-1470 | 0.0172 |

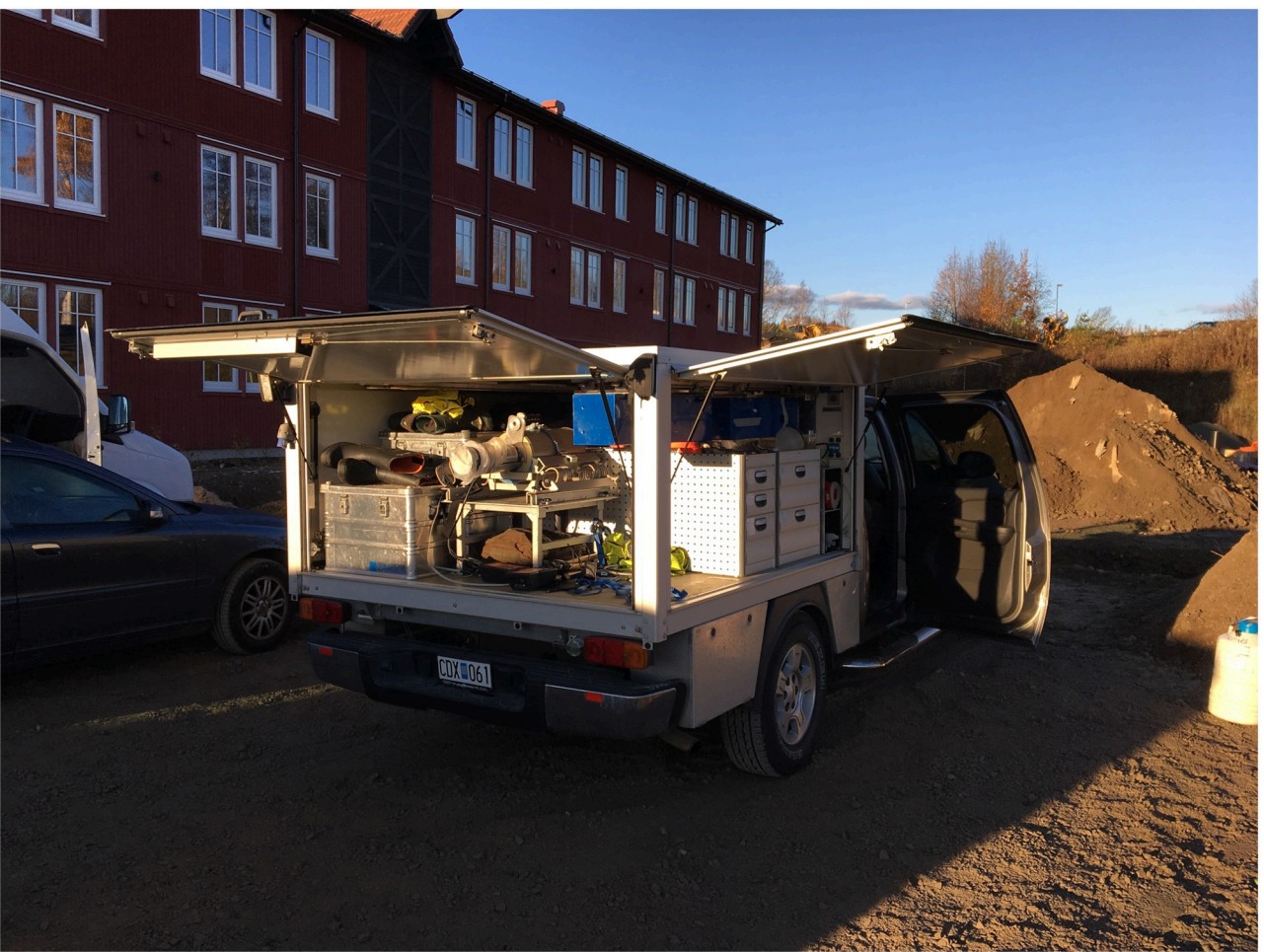

**Fig 4. A picture of the open flat bed portion of the mobile gamma spectrometry vehicle used in the experiment.** The 123% HPGe detector with it's dewar can be seen in the back of the flat bed, with the active medium of the detector pointing to the back of the vehicle. The two blue boxes in the top of the right side of the flat bed are the 4l NaI(Tl) detectors. The rear pillars of the flat bed structure can be also clearly seen.

in the efficiency at 90 and 270 degrees, most probably due to the structural pillars at the back of the custom flatbed of the vehicle (can be seen in the Fig 4). Thus, it was chosen to normalize the relative efficiency to the 80 degree value to avoid the relative efficiency being much higher than 1 and to stay true to the calibration performed.

Resulting relative angular efficiencies normalized to their respective values are displayed in Fig 5 for $^{60}$Co and $^{137}$Cs. Reference efficiency values measured at 90 degrees from the detectors are given in the Table 2.

For the variations of the algorithm using fixed counting efficiency (HPGe90, NaI90 and Mult90), the efficiency was obtained by using a relative efficiency value of 90 degrees for all of the angles.

### Simulated data

The simulated data was calculated by taking a random sample from a Poisson distribution, with $\lambda_i$ of the Poisson distribution representing the mean number of counts in a detector at a measurement point $i$ per unit of time $t$ [16]. To be able to perform a direct comparison with

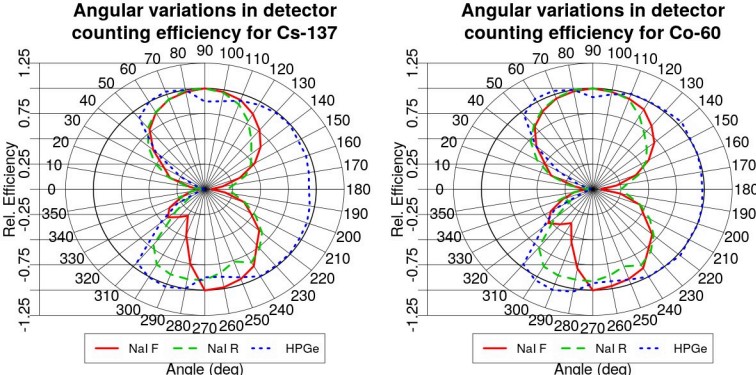

**Fig 5. Polar plots of relative angular variations of counting efficiency for front 4l NaI detector (red), rear 4l NaI detector (green) and HPGe detector (blue) for $^{137}$Cs (left) and $^{60}$Co (right).** The relative efficiency of both NaI detectors were normalised to their respective efficiency value at 90 degrees, while at 80 degrees for the HPGe detector. Angle of 0 degrees represents the driving direction of the vehicle.

experimental data in the upcoming study, measurement point coordinates have to be comparable. Thus, it was chosen to take the GPS coordinates of one complete pass of the experimental passes past the sources as the main coordinates, and to calculate the simulated data for these measurement points. Background count rate $c$ per unit time $t$ for each radionuclide and individual detector were evaluated as a mean value of counts in the selected ROI in the gamma spectra while driving back and forth along the predetermined path (Fig 2) in absence of any radioactive sources. The background values used for calculating the simulated data are given in the Table 3. Mean number of counts at every measurement position $i$ were calculated by applying the activity and position of the source corresponding to a selected experimental setup (Table 1) and average background level in the area $c$ to the equation Eq 8. Angular variations in the counting efficiencies of primary gamma photons (Fig 5) for all three detectors were included in the calculation of the simulated data alongside the individual offset of each detector from the position of the GPS antenna on the vehicle. Since the Bayesian algorithm is sensitive to the shape of the peak in the measurement time-series, total number of 10 Poisson distributed simulated values (realizations) of number of counts in the detectors for each type of the source in source setups were calculated to make the evaluation of average performance of the algorithm possible. The modeled data series, (matrix $Z$ in Eq 9), was then used as an input into the Bayesian algorithm.

To be able to compare the discrepancies in the results easier, signal-to-noise ratios (SNR) were calculated for all individual setup and source combinations. SNRs were calculated as a ratio between the maximum amplitude of signal and the standard deviation of the noise—variations in the counts due to background radiation. Henceforth, the following formula was then used in the calculations:

$$SNR = \frac{I - \bar{B}}{\sqrt{\bar{B}}}, \tag{12}$$

**Table 3. Average background counts in the respective ROIs in the detectors per 1 second during the mobile gamma spectrometry experiment.**

| ROI | HPGe | Front NaI(Tl) | Rear NaI(Tl) |
|---|---|---|---|
| $^{137}$Cs | 0.35 | 65.0 | 53.8 |
| $^{60}$Co | 0.25 | 65.1 | 45.8 |

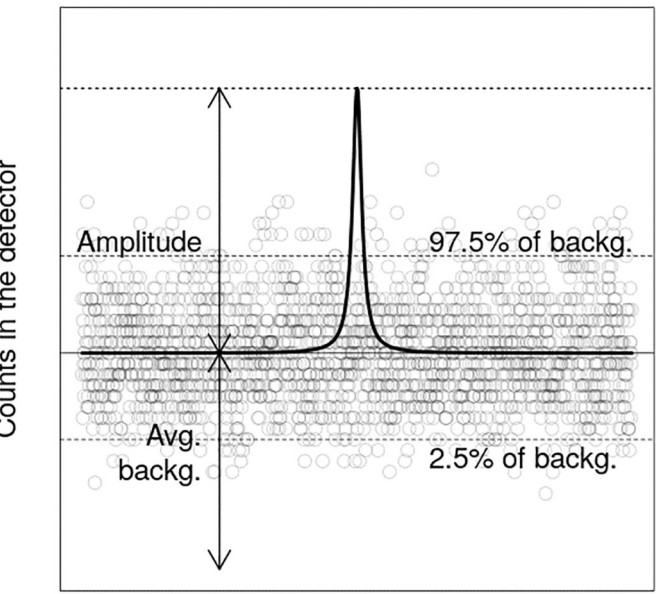

**Fig 6. Graphical representation of the SNR calculation.** Black line represents the mean value of counts in the detector due to the source and background radiation. Black transparent circles represents one realization of counts in the detector from the background for the set of measurements. Thin dashed lines represent the 2.5 and 97.5% of the background counts distribution. Thicker dashed line marks the maximum number of counts. The SNR is calculated by dividing the amplitude by the standard deviation of background distribution, which is the square root of the average background counts.

where $I - \bar{B}$ corresponds to the maximum, average background subtracted, amplitude of the peak in the count rate time-series and $\bar{B}$ is the average number of background counts in the detector. Graphical representation of SNR is displayed in Fig 6. Obtained SNR values for all of the individual detectors and their means for both sources are displayed in Table 4.

## Bayesian algorithm

The algorithm generating samples from the posterior distribution through a Markov Chain Monte Carlo (MCMC) algorithm was modified to be able to utilize the data from all three of the detectors present in the vehicle simultaneously (two 4l NaI(Tl) and one 123% HPGe) in the Bayesian inference [14, 17]. Due to significant deviations ($> 2$ m) of the detectors' actual positions from the position of GPS antenna, positions of the detectors were corrected based on the heading of the vehicle and the measured relative offset of the detectors' position from the GPS receiver. Angular variations in the counting efficiencies for the primary gamma photons were introduced for all of the detectors present in the vehicle (as described in Eq 4). Due to the differences in the detectors, background parameters $c_j$ (Eq 9) were estimated for each detector $j$ alongside the position and activity of the source. The initial coordinate for starting the MCMC algorithm was chosen to be the measurement coordinate where counts in the detector were the highest to have a good initial proposal for the MCMC algorithm. As detectors used in the vehicle are incapable of recording the angle of incidence of the incident photon, the general solution to the source location problem is therefore a bivariate distribution. The fact that the starting coordinate is in the middle of two local maxima of the posterior distribution for the

**Table 4. SNR values of the theoretical data set for all three detectors—HPGe, front (F) NaI, rear (R) NaI and two sources with their respective mean and standard deviation (S.D.) values for $^{137}$Cs and $^{60}$Co.**

| Setup number | $^{137}$Cs | | | | | $^{60}$Co | | | | |
|---|---|---|---|---|---|---|---|---|---|---|
| | HPGe | NaI(F) | NaI(R) | Mean | S.D. | HPGe | NaI(F) | NaI(R) | Mean | S.D. |
| 1 | 33.20 | 27.32 | 29.97 | 30.16 | 2.94 | 44.76 | 23.08 | 33.36 | 33.73 | 10.84 |
| 2 | 19.83 | 15.94 | 15.66 | 17.14 | 2.33 | 11.99 | 7.35 | 9.20 | 9.51 | 2.34 |
| 3 | 6.81 | 5.26 | 6.06 | 6.04 | 0.78 | 5.39 | 3.52 | 3.93 | 4.28 | 0.98 |
| 4 | 2.26 | 1.82 | 2.07 | 2.05 | 0.22 | 2.28 | 1.43 | 1.67 | 1.79 | 0.44 |
| 5 | 22.13 | 17.25 | 19.83 | 19.74 | 2.44 | 68.35 | 32.33 | 49.83 | 50.17 | 18.01 |
| 6 | 7.79 | 6.02 | 6.74 | 6.85 | 0.89 | 26.39 | 14.73 | 19.02 | 20.05 | 5.90 |
| 7 | 3.95 | 3.39 | 3.42 | 3.59 | 0.32 | 6.05 | 3.69 | 4.54 | 4.76 | 1.20 |
| 8 | 1.31 | 1.03 | 1.16 | 1.17 | 0.14 | 3.36 | 2.03 | 2.39 | 2.59 | 0.69 |
| 9 | 25.92 | 19.78 | 22.79 | 22.83 | 3.07 | 21.34 | 13.17 | 14.68 | 16.40 | 4.35 |
| 10 | 11.72 | 9.19 | 10.43 | 10.45 | 1.27 | 9.40 | 5.88 | 6.88 | 7.39 | 1.81 |
| 11 | 4.37 | 3.53 | 4.01 | 3.97 | 0.42 | 4.01 | 2.51 | 2.82 | 3.11 | 0.79 |
| 12 | 2.13 | 1.74 | 1.80 | 1.89 | 0.21 | 1.61 | 1.00 | 1.24 | 1.28 | 0.31 |
| 13 | 19.99 | 16.00 | 17.27 | 17.75 | 2.04 | 15.53 | 9.82 | 12.15 | 12.50 | 2.87 |
| 14 | 7.56 | 6.30 | 6.49 | 6.78 | 0.68 | 6.29 | 4.04 | 4.98 | 5.10 | 1.13 |
| 15 | 3.31 | 2.65 | 2.76 | 2.91 | 0.35 | 3.37 | 2.12 | 2.62 | 2.70 | 0.63 |
| 16 | 1.58 | 1.26 | 1.42 | 1.42 | 0.16 | 1.94 | 1.22 | 1.37 | 1.51 | 0.38 |
| 17 | 29.38 | 24.78 | 26.09 | 26.75 | 2.37 | 12.17 | 7.94 | 9.80 | 9.97 | 2.12 |
| 18 | 10.00 | 8.07 | 9.08 | 9.05 | 0.97 | 7.11 | 4.57 | 5.13 | 5.60 | 1.33 |
| 19 | 4.49 | 3.70 | 4.17 | 4.12 | 0.40 | 3.78 | 2.38 | 2.68 | 2.95 | 0.74 |
| 20 | 2.40 | 1.86 | 2.11 | 2.12 | 0.27 | 2.11 | 1.35 | 1.52 | 1.66 | 0.40 |

source position, there is a tendency of the MCMC algorithm to stay within the side of the vehicle direction to which the first jump was made. Irrespective of whether there is additional information (arising due to angular variations of the counting efficiency, detector position offsets, etc.) or not, the algorithm will most probably stay on the side of the road/vehicle direction chosen with the first step. This constrains the ability of the algorithm to explore the full bivariate distribution correctly. To mitigate this behaviour, the position of the proposal coordinate is mirrored to the opposite side of the road each 1000 MCMC iterations.

Inference was performed for 10 different data realizations (described previously) for each source setup and radionuclide. Bayesian algorithm was running for 30000 iterations with the burn-in of 10000 iterations. Maximum aposteriori (MAP) values—values with the highest probability—of the posterior distributions were extracted for activity and position of the source for each data realization, source setup and source type. These MAP values are further regarded as Bayesian estimations (or simply estimations).

A "position estimation deviation" (PED) was defined as a distance between the MAP position and the actual position of the source. Due to the bivariate nature of the posterior distribution for source position, the estimated position can be on either side of the road if there is not enough information in the data for the algorithm to choose a particular side of the road. In a case where the estimated position is on the other side of the road than the actual position of the source, the PED will indicate a falsely high deviation (close to two times the distance from the source to the road). To correct this a "merged MAP position" (MMAPP) was calculated— the estimated positions of the sources were mirrored along the trajectory of the mobile system for estimations, where estimated position was on the other side of the trajectory than the source was actually positioned. Then, the PED values were recalculated as a distance between the MMAPP and the actual position of the source.

To be able to compare the results obtained using different variations of the algorithm easier, relative deviations were calculated for activity and position of the source. Activity relative deviation (ARD) was expressed as a relative difference between the estimated and the actual activity in percent, thus negative ARD values indicate underestimation of the activity values, while positive—overestimation. Position relative deviation (PRD) was calculated as a ratio of a PED value and the distance from the source to the road side. PRD, thus can be regarded as the distance between the actual and estimated source position, expressed in the percent of the distance from the source to the road.

As 10 different Bayesian estimations were calculated for the different realizations of the data, there were 10 corresponding combinations of ARD and PRD values for the same conditions (experimental setup number, radionuclide type and variation of the algorithm). The ARD and PRD values for the same conditions, were then analyzed as probability distributions. Thus, 2.5%, 50% (median) and 97.5% quantiles were calculated for the ARD and PRD distributions. Medians of the distributions together with the 95% range of the distributions around the median will to a large extent display the differences in accuracy and precision of the Bayesian estimations for different variations and conditions of the algorithm.

Furthermore, median values of the ARD and PRD distributions throughout the experimental setups for a given variation of the algorithm and radionuclide were analyzed as distributions, and will be further denoted as ARRD and PRRD distributions respectively. Medians, 2.5% and 97.5% quantiles alongside the 97.5%-2.5% interquantile distance (further referred to as ID) were evaluated for the ARDD and PRDD distributions to obtain the average accuracy and precision of the variations of the algorithm in more detail.

ARD and PRD distributions has a scope of one particular experimental setup. ARDD and PRDD distributions has a scope over all of the experimental setups. Thus, ARD and PRD distributions displays how well the given variation of the algorithm could predict the activity and position for that particular experimental setup and radionuclide, while ARDD and PRDD distributions displays how well the algorithm was performing in estimating the activity and position of the source for a given variation of the algorithm for all of the experimental setups on average.

## Results and discussion

Median values and 2.5% and 97.5% quantiles of the obtained ARD (plotted with activity on the y axis rather than percent for easier analysis) and PRD distributions are displayed graphically in Figs 7 and 8 for $^{137}$Cs and $^{60}$Co respectively. Median values of ARD and PRD distributions for every combination of experimental source setup, algorithm variation and radionuclide type are displayed in Tables 5 and 6 alongside the SNR of the data.

Due to the nature of the design of the experiment (variable SNRs approaching the detection limit), it was possible to evaluate theoretical "threshold" SNR levels, where the uncertainty in the estimations started to increase. In Fig 9, the medians of ARD and PRD distributions for different SNRs are plotted with SNR on the x-axis. For the variations of algorithm using data from HPGe, NaI and multiple detectors, thresholds were evaluated graphically from combinations of ARD and PRD graphs. Obtained approximate threshold levels are 8, 5 and 3 for the variations of the algorithm using only HPGe data, only NaI data and using data from multiple detectors respectively. For all of the estimations with data SNR above the individual threshold level, the estimations are reasonably close to the true values (majority of absolute ARD and PRD values are less than 50%).

Unsurprisingly—the bigger the combined counting efficiency of the detector system, the lower the threshold above which the estimations start to deviate strongly from the actual

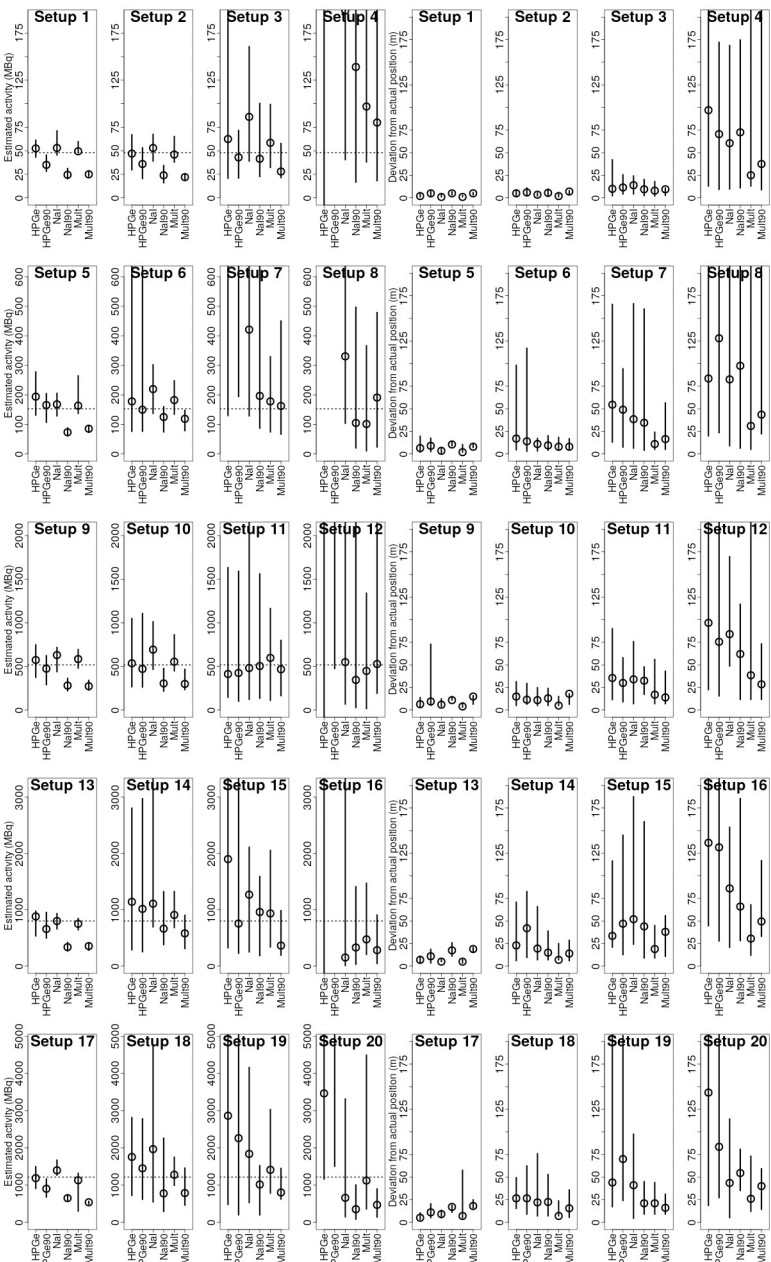

**Fig 7. Quantile plots of the activity MAP (left) and position estimation deviation (right) distributions for $^{137}$Cs.** The quantiles displayed are 2.5%, 50% and 97.5% quantiles for estimations obtained using the six different variations of the Bayesian algorithm. The empty dot marks the median of the distribution. The span of the line marks the 95% quantile range around the median of the distribution. The dotted line in the graphs on the left marks the actual source activity. The y scale is not adjusted to fit all of the points due to low relative importance of far-away points.

activity and position of the source. Obtained relationship of threshold SNR levels and combined counting efficiencies for different variations of the algorithm are displayed in Fig 10. Thus, to be able to locate weaker sources positioned further away in practice more data needs to be collected. To achieve that either total efficiency of the system could be increased by employing a bigger number of more efficient detectors or the exposure time increased by

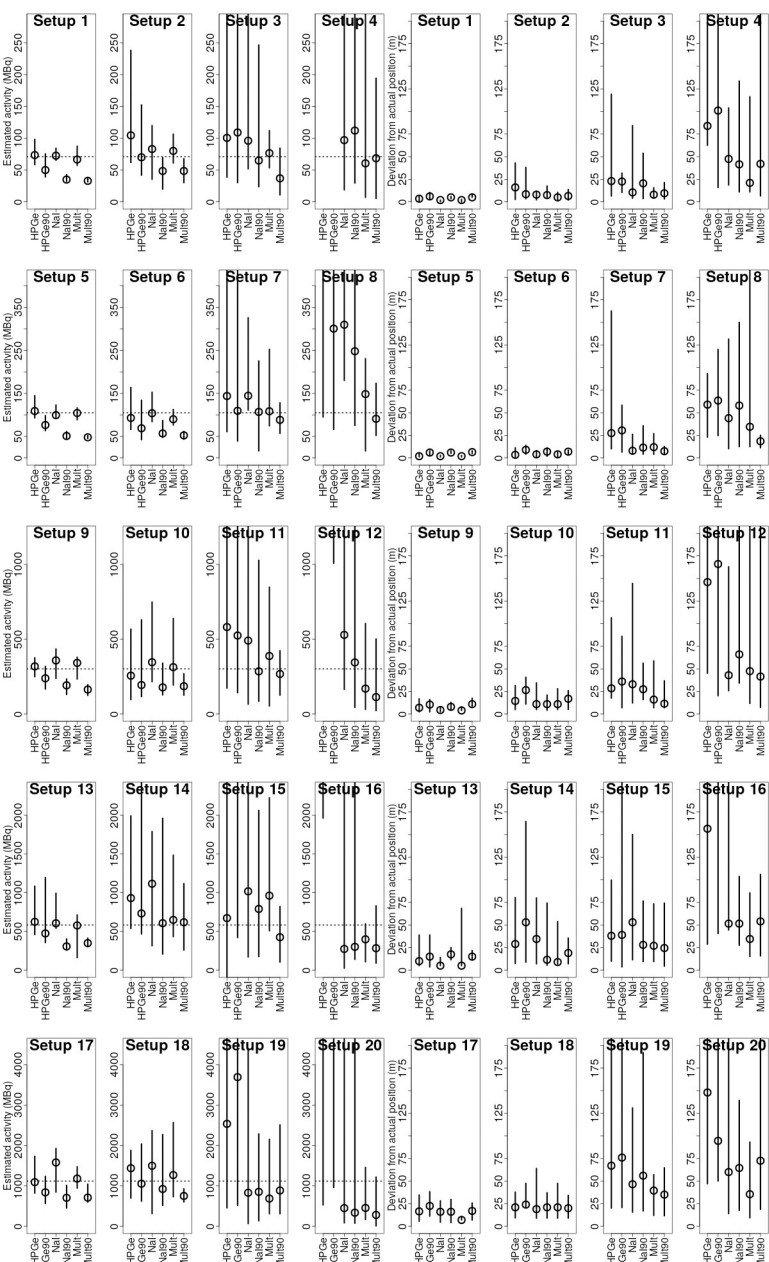

**Fig 8. Quantile plots of the activity MAP (left) and position estimation deviation (right) distributions for $^{60}$Co.** The quantiles displayed are 2.5%, 50% and 97.5% quantiles for estimations obtained using the six different variations of the Bayesian algorithm. The empty dot marks the median of the distribution. The span of the line marks the 95% quantile range around the median of the distribution. The dotted line in the graphs on the left marks the actual source activity. The y scale is not adjusted to fit all of the points due to low relative importance of far-away points.

performing multiple passes past the source (alternatively—performing a pass at a lower speed). Both of the options would result in more collected data and lower SNR threshold.

From Tables 5 and 6, it is evident that on average, estimations obtained using all individual variations of the algorithm do not deviate more than 50% from the actual activity and more then 40% from the actual position in terms of the distance from the source to the road. Maximum median values of the ARDD distributions were 36% and 39% for $^{137}$Cs and $^{60}$Co

**Table 5. Calculated median values of ARD distributions for all of the source setups and algorithm variations.**

| Setup number | Activity relative deviation (%) | | | | | | | | | | | | SNR | |
|---|---|---|---|---|---|---|---|---|---|---|---|---|---|---|
| | HPGe | | HPGe90 | | NaI | | NaI90 | | Mult | | Mult90 | | Mean | |
| | $^{137}$Cs | $^{60}$Co | $^{137}$Cs | $^{60}$Co | $^{137}$Cs | $^{60}$Co | $^{137}$Cs | $^{60}$Co | $^{137}$Cs | $^{60}$Co | $^{137}$Cs | $^{60}$Co | $^{137}$Cs | $^{60}$Co |
| 1 | 8 | 4 | -25 | -30 | 10 | 1 | -50 | -51 | 4 | -7 | -48 | -54 | 30.2 | 33.7 |
| 2 | -2 | 46 | -25 | -1 | 10 | 17 | -50 | -32 | -4 | 13 | -54 | -32 | 17.1 | 9.5 |
| 3 | 29 | 41 | -10 | 54 | 79 | 35 | -12 | -8 | 21 | 7 | -42 | -48 | 6 | 4.3 |
| 4 | 464000 | 3900000 | 295000 | 273000 | 315 | 37 | 190 | 58 | 102 | -15 | 67 | -4 | 2 | 1.8 |
| 5 | 27 | 4 | 8 | -28 | 10 | -5 | -52 | -51 | 7 | -1 | -44 | -54 | 19.7 | 50.2 |
| 6 | 16 | -11 | -2 | -34 | 44 | -1 | -18 | -46 | 19 | -14 | -22 | -50 | 6.8 | 20.1 |
| 7 | 1789 | 37 | 716 | 5 | 175 | 37 | 28 | 2 | 16 | 3 | 6 | -16 | 3.6 | 4.8 |
| 8 | 8370000 | 542 | 184000 | 186 | 116 | 195 | -31 | 136 | -33 | 41 | 25 | -13 | 1.2 | 2.6 |
| 9 | 11 | 5 | -8 | -21 | 22 | 19 | -46 | -36 | 13 | 13 | -47 | -46 | 22.8 | 16.4 |
| 10 | 3 | -15 | -9 | -36 | 34 | 15 | -41 | -41 | 7 | 4 | -42 | -38 | 10.4 | 7.4 |
| 11 | -20 | 93 | -18 | 74 | -7 | 63 | -3 | -5 | 15 | 28 | -10 | -11 | 4 | 3.1 |
| 12 | 448 | 72800 | 5200 | 57500 | 6 | 75 | -33 | 14 | -14 | -44 | 2 | -62 | 1.9 | 1.3 |
| 13 | 10 | 7 | -18 | -19 | 0 | 4 | -58 | -48 | -6 | -1 | -56 | -40 | 17.8 | 12.5 |
| 14 | 42 | 60 | 27 | 26 | 38 | 91 | -17 | 4 | 13 | 11 | -27 | 6 | 6.8 | 5.1 |
| 15 | 137 | 15 | -6 | 356 | 58 | 75 | 20 | 36 | 17 | 65 | -54 | -27 | 2.9 | 2.7 |
| 16 | 14600 | 17600 | 12500 | 6300 | -81 | -53 | -59 | -48 | -41 | -32 | -65 | -51 | 1.4 | 1.5 |
| 17 | -2 | -2 | -26 | -25 | 15 | 41 | -47 | -37 | -7 | 6 | -56 | -37 | 26.8 | 10 |
| 18 | 45 | 29 | 19 | -6 | 62 | 34 | -36 | -5 | 5 | 13 | -35 | -33 | 9.1 | 5.6 |
| 19 | 136 | 127 | 86 | 230 | 51 | -26 | -16 | 1 | 16 | -39 | -34 | -21 | 4.1 | 3 |
| 20 | 185 | 2790 | 3480 | 2300 | -46 | -60 | -71 | -80 | -8 | -59 | -61 | -75 | 2.1 | 1.7 |
| 2.5% | -11 | -13 | -26 | -35 | -64 | -57 | -65 | -66 | -37 | -52 | -63 | -69 | | |
| Median | 36 | 39 | 3 | 16 | 28 | 26 | -34 | -20 | 7 | 4 | -42 | -38 | | |
| 97.5% | 4610000 | 2080000 | 243000 | 171000 | 248 | 146 | 113 | 99 | 64 | 54 | 47 | 1 | | |
| ID | 4610000 | 2080000 | 243000 | 171000 | 313 | 202 | 178 | 165 | 101 | 105 | 110 | 70 | | |

Variations of the algorithm from the left side—using data from only HPGe with angular variations in counting efficiency (marked as HPGe), only HPGe fixed counting efficiency (HPGe90), only the rear NaI detector with angular variations in counting efficiency (NaI), only the rear NaI detector with fixed counting efficiency (NaI90), using data from all three of the detectors present in the vehicle in conjuction with the angular variations of efficiency (Mult) and using data from all three of the detectors present in the vehicle in conjunction with fixed counting efficiency (Mult90). In the bottom part of the table, quantiles of the distributions of median values themselves throughout the experimental setups (ARDD distributions) are displayed. Mean SNRs from Table 4 are also displayed on the right part of the table for reference convenience.

respectively obtained using the HPGe variation of the algorithm. Minimum—-42% and -38% for $^{137}$Cs and $^{60}$Co respectively using Mult90 variation of the algorithm. Similarly, the highest median value for the PRDD distributions is 37% and 36% for $^{137}$Cs and $^{60}$Co respectively obtained with HPGe90 variation of the algorithm. And the minimum median values for the PRDD distributions were 14% and 16% for the sources respectively for the Mult variation of the algorithm. Biggest absolute values of medians of ARDD and PRDD distributions, marking the average accuracy of activity and position estimation, were 42% and 37%. On average, the activity estimates are expected to be less than 50% off from the actual value, for SNR-values ranging from 1.5 to 50. The corresponding value for the position estimates is less than 40% for all variations of the algorithm. For the most advanced version of the algorithm (incorporating angular variations in counting efficiency and data from multiple detectors), the expected average deviation from the actual activity is less than 10% ± 55% and less than 20% ± 20% from the actual position (in terms of source to road distance).

**Table 6. Calculated median values of PRD distributions for all of the source setups and algorithm variations.**

| Setup number | Position relative deviation(%) | | | | | | | | | | | | | SNR | |
|---|---|---|---|---|---|---|---|---|---|---|---|---|---|---|---|
| | HPGe | | HPGe90 | | NaI | | NaI90 | | Mult | | Mult90 | | | Mean | |
| | $^{137}$Cs | $^{60}$Co | $^{137}$Cs | $^{60}$Co | $^{137}$Cs | $^{60}$Co | $^{137}$Cs | $^{60}$Co | $^{137}$Cs | $^{60}$Co | $^{137}$Cs | $^{60}$Co | | $^{137}$Cs | $^{60}$Co |
| 1 | 17 | 18 | 50 | 27 | 8 | 9 | 42 | 23 | 8 | 9 | 42 | 23 | | 30.2 | 33.7 |
| 2 | 23 | 50 | 27 | 25 | 18 | 25 | 27 | 25 | 9 | 16 | 32 | 19 | | 17.1 | 9.5 |
| 3 | 31 | 44 | 38 | 42 | 44 | 19 | 31 | 38 | 25 | 15 | 31 | 19 | | 6 | 4.3 |
| 4 | 187 | 117 | 135 | 140 | 115 | 67 | 138 | 58 | 48 | 29 | 73 | 58 | | 2 | 1.8 |
| 5 | 19 | 9 | 28 | 27 | 12 | 9 | 31 | 27 | 6 | 9 | 25 | 27 | | 19.7 | 50.2 |
| 6 | 33 | 12 | 27 | 28 | 21 | 12 | 19 | 22 | 15 | 12 | 15 | 22 | | 6.8 | 20.1 |
| 7 | 75 | 54 | 68 | 58 | 53 | 15 | 47 | 23 | 15 | 23 | 22 | 15 | | 3.6 | 4.8 |
| 8 | 82 | 82 | 125 | 89 | 80 | 61 | 96 | 81 | 30 | 47 | 43 | 25 | | 1.2 | 2.6 |
| 9 | 12 | 13 | 19 | 19 | 12 | 8 | 21 | 15 | 8 | 8 | 29 | 21 | | 22.8 | 16.4 |
| 10 | 21 | 19 | 17 | 36 | 15 | 15 | 18 | 15 | 7 | 15 | 25 | 24 | | 10.4 | 7.4 |
| 11 | 35 | 27 | 29 | 35 | 33 | 32 | 31 | 27 | 17 | 16 | 14 | 12 | | 4 | 3.1 |
| 12 | 73 | 111 | 58 | 126 | 64 | 33 | 47 | 50 | 29 | 36 | 21 | 32 | | 1.9 | 1.3 |
| 13 | 10 | 14 | 15 | 21 | 7 | 7 | 25 | 25 | 7 | 7 | 26 | 21 | | 17.8 | 12.5 |
| 14 | 23 | 28 | 41 | 52 | 20 | 33 | 15 | 12 | 7 | 9 | 14 | 19 | | 6.8 | 5.1 |
| 15 | 26 | 29 | 36 | 30 | 39 | 40 | 33 | 21 | 14 | 20 | 29 | 18 | | 2.9 | 2.7 |
| 16 | 84 | 96 | 81 | 144 | 53 | 32 | 41 | 32 | 19 | 21 | 31 | 33 | | 1.4 | 1.5 |
| 17 | 7 | 16 | 15 | 22 | 12 | 16 | 24 | 16 | 10 | 7 | 25 | 17 | | 26.8 | 10 |
| 18 | 25 | 16 | 25 | 18 | 22 | 14 | 22 | 16 | 7 | 16 | 16 | 15 | | 9.1 | 5.6 |
| 19 | 33 | 41 | 53 | 47 | 31 | 28 | 16 | 59 | 16 | 25 | 12 | 22 | | 4.1 | 3 |
| 20 | 89 | 77 | 52 | 49 | 27 | 31 | 33 | 40 | 16 | 19 | 25 | 38 | | 2.1 | 1.7 |
| 2.5% | 8 | 10 | 15 | 18 | 7 | 7 | 15 | 13 | 6 | 7 | 13 | 13 | | | |
| Median | 28 | 28 | 37 | 36 | 24 | 22 | 31 | 25 | 14 | 16 | 25 | 22 | | | |
| 97.5% | 140 | 114 | 130 | 142 | 98 | 64 | 118 | 71 | 39 | 42 | 59 | 48 | | | |
| ID | 132 | 104 | 115 | 124 | 91 | 57 | 103 | 57 | 33 | 35 | 46 | 35 | | | |

Variations of the algorithm from the left side—using data from only HPGe with angular variations in counting efficiency (marked as HPGe), only HPGe fixed counting efficiency (HPGe90), only the rear NaI detector with angular variations in counting efficiency (NaI), only the rear NaI detector with fixed counting efficiency (NaI90), using data from all three of the detectors present in the vehicle in conjuction with the angular variations of efficiency (Mult) and using data from all three of the detectors present in the vehicle in conjunction with fixed counting efficiency (Mult90). In the bottom part of the table, quantiles of the distributions of median values themselves throughout the experimental setups (ARDD distributions) are displayed. Mean SNRs from Table 4 are also displayed on the right part of the table for reference convenience.

It can be observed that while the Table 6 and especially Table 5 indicate the average performance of the variation of the algorithm for the selected source quite well, there are some extremely high estimated values for the experimental setups where the SNR was the lowest—e.g. median of the estimated activities for $^{137}$Cs in experimental setup 4 for HPGe variation of the algorithm was 464000% when the SNR was 2—decreasing the average performance estimate of the algorithm. Such a result basically displays that the algorithm had no idea where and what the activity of the source is. And while the average performance of the algorithm throughout diferent SNR situations is also of interest, this does not completely reflect the possibilities of the algorithm. To get a better picture on the performance of the algorithm, a new Table 7 was compiled from the rows with the best SNRs from Table 5 (The data is also visualised in a boxplot form in Fig 11). In this Table 7, it can be clearly seen that the variations of the algorithm using the angular efficiency variations (HPGe, NaI and Mult) were very close to the actual activity values, if had a tendency to slightly overestimate the activities, with median

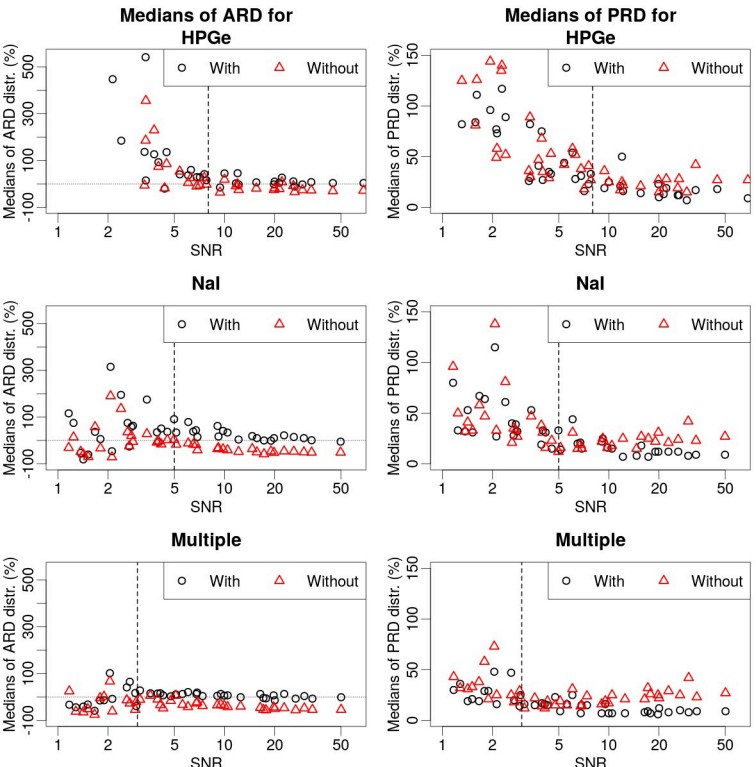

**Fig 9. Medians of ARD (left) and PRD (right) distributions plotted against SNR values for respective experimental setups and radionuclides.** Variations of the algorithm using different data were separated in to different graphs (HPGe, NaI, Multiple), with and without the use of angular dependence in Eq 9. Horizontal line in ARD graphs marks the 0% relative deviation—actual source activity. Vertical line in all of the graphs marks the approximate individual threshold level for the variations of the algorithm using specific type of the data.

values throughout the data with best SNRs covering a range from -1% to 10%. On the other hand, the variations of the algorithm not using the angular variations of the efficiency (HPGe90, NaI90, Mult90) had a strong tendency to underestimate the activity of the source with median values varying from -18% to -50%. This underestimation could be explained by the difference in detector response with and without angular variations of efficiency.

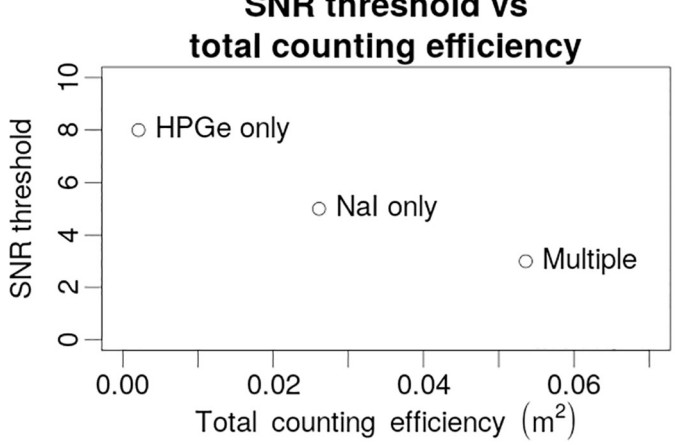

**Fig 10. Relationship between the SNR threshold and total counting efficiency of the system for different variations of the algorithm.**

**Table 7. A subset of Table 5, displaying only the results from setups with the highest SNRs.**

| Setup number | HPGe | | HPGe90 | | NaI | | NaI90 | | Mult | | Mult90 | | SNR | |
|---|---|---|---|---|---|---|---|---|---|---|---|---|---|---|
| | $^{137}$Cs | $^{60}$Co | $^{137}$Cs | $^{60}$Co | $^{137}$Cs | $^{60}$Co | $^{137}$Cs | $^{60}$Co | $^{137}$Cs | $^{60}$Co | $^{137}$Cs | $^{60}$Co | $^{137}$Cs | $^{60}$Co |
| 1 | 8 | 4 | -25 | -30 | 10 | 1 | -50 | -51 | 4 | -7 | -48 | -54 | 30.2 | 33.7 |
| 5 | 27 | 4 | 8 | -28 | 10 | -5 | -52 | -51 | 7 | -1 | -44 | -54 | 19.7 | 50.2 |
| 9 | 11 | 5 | -8 | -21 | 22 | 19 | -46 | -36 | 13 | 13 | -47 | -46 | 22.8 | 16.4 |
| 13 | 10 | 7 | -18 | -19 | 0 | 4 | -58 | -48 | -6 | -1 | -56 | -40 | 17.8 | 12.5 |
| 17 | -2 | -2 | -26 | -25 | 15 | 41 | -47 | -37 | -7 | 6 | -56 | -37 | 26.8 | 10 |
| 2.5% | -1 | -1.4 | -25.9 | -29.8 | 1 | -4.4 | -57.4 | -51 | -6.9 | -6.4 | -56 | -54 | 17.99 | 10.25 |
| Median | 10 | 4 | -18 | -25 | 10 | 4 | -50 | -48 | 4 | -1 | -48 | -46 | 22.8 | 16.4 |
| 97.5% | 25.4 | 6.8 | 6.4 | -19.2 | 21.3 | 38.8 | -46.1 | -36.1 | 12.4 | 12.3 | -44.3 | -37.3 | 29.86 | 48.55 |

In the bottom of the table, 2.5% and 97.5% quantiles and medians of the medians of ARD distributions for a particular variation of the algorithm and radionuclide are displayed.

Because the relative angular efficiencies were normalized to 90 and 80 degree values for NaI and HPGe detectors respectively, the maxima of the relative angular efficiencies are positioned at 90 and 80 degree points also, respectively. Due to the relative symmetry of the relative angular efficiencies (the response is somewhat distorted for the left side of NaI detectors due to their position being very close to the right side of the vehicle), the relative efficiency values on the left side of the vehicle are similar. If this distortion for the NaI detector response could be ignored, the relative efficiency then resembles basically a figure-of-eight pattern, due to the fact that the crystal is elongated along the longitudinal axis of the vehicle, and has a bigger surface area towards the source at close to 90 and 270 degrees. The efficiency then gradually decreases until the minima points at 0 and 180 degrees. Now, if a detector with such a response would pass a source at a certain distance, the relative efficiency would increase, reach a maximum at the closest distance to the source, and then start to decrease once again. This

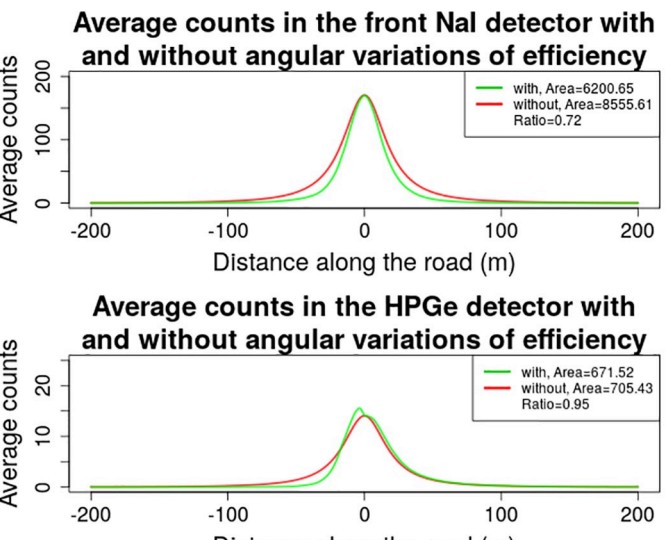

**Fig 11. Boxplot of ARD values for different variations of the algorithms and different sources throughout the experimental setups 1, 5, 9, 13 and 17.** On the right, the SNR box plot for the aforementioned SNRs of the data is displayed.

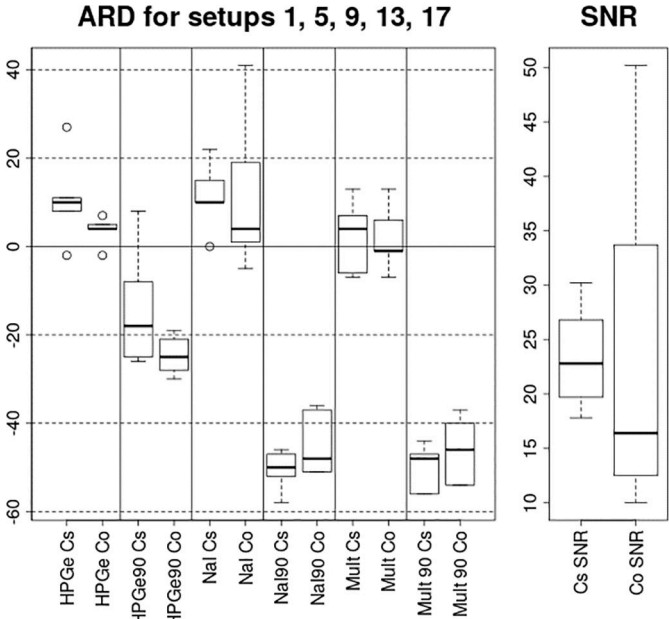

**Fig 12. Comparison of average number of counts in NaI and HPGe detectors, while traveling past a radioactive source positioned some distance away from the road at position 0, using angular variations of efficiency and using a single, fixed value for the counting efficiency of the detector.**

behaviour is correctly taken into account for the variations of the algorithm using angular variations of the efficiency. But, if the efficiency is kept fixed at e.g. 90 degree values, then the estimated efficiency for all of the angles of incidence except 90 degrees, will be higher than the actual efficiency. The situation is also very similar for the HPGe detector. Due to the dewar and other structural parts of the detector/vehicle obscuring the line-of-sight to the detector from about 310 to 50 degrees, the efficiency is extremely low, basically blocking all of the photons from reaching the detector. By having a fixed counting efficiency value, there are still significant photons expected in this region of angles of incidence. This is visualised graphically in Fig 12.

Then, if the relationship between the activity, counting efficiency and the number of counts would be considered, $A \propto N/\varepsilon$, a relative increase in the efficiency of the detector, by comparing to the actual efficiency value for that angle of incidence, would suggest an increase in the expected number of counts in the detector. As the data was simulated using the angular variations of the efficiency, the estimated activity thus has to be reduced to get a better fit of the data.

Although theoretical results look promising, real-world performance of such multiple detector Bayesian-based system might be affected by multiple additional factors. Gain drift due to temperature variations in photo multiplier (PM) tube of the NaI detectors could significantly distort the distribution of the count rate in the detector. Also, due to the sensitivity of the algorithm to the shape of the peak in the count rate time-series, the algorithm is very susceptible to any inconsistencies within sampling times of GPS position and the time-stamping of the measurements. Thus, actual improvements in the estimations produced using real-world data should be studied further. Furthermore, applicability of such Bayesian approach could also be tested on neutron detector systems using a proper neutron detection model.

## Conclusions

Based on the simulated data, the ability of all six individual variations of the algorithm to estimate the activity and position of the source is acceptable for use in one-pass emergency situations performed at speeds lower than 50 km/h.

Estimation precision and reliability of the Bayesian algorithms depends mainly on the signal-to-noise (SNR) of the data. The bigger the combined efficiency of the system, the lower the SNR threshold of the system, the more reliable and precise the estimations will be in situations when the source is near the detection limit. Alternatively, if there is no time constraint on the survey, additional passes could be performed to increase the amount of information collected for the same SNR level of the data.

Inclusion of the angular variations in the counting efficiency of the detectors in the Bayesian model allows to take in to account the changing efficiency of the detector with changing angle of incidence of the photons, which corrects the underestimation of the activities. Despite that, in emergency situations where such calibration is not possible, fixed values for counting efficiency of the detectors can provide with similar results if calibrated accordingly. Additionally, the Bayesian inference would be performed faster due to less code in the Markov Chain Monte-Carlo loop.

## Supporting information

**S1 Data.**
(GZ)

## Acknowledgments

The authors thank Mats Hansson, Mattias Jönsson and Kurt Sundin for assisting with the measurements and Barsebäck Kraft AB, Barsebäck Gods AB and Löddeköpinge Rescue Services for access to areas for carrying out parts of the measurements.

## Author Contributions

**Conceptualization:** Antanas Bukartas, Jonas Wallin, Robert Finck, Christopher Rääf.

**Data curation:** Antanas Bukartas.

**Formal analysis:** Antanas Bukartas.

**Funding acquisition:** Robert Finck, Christopher Rääf.

**Investigation:** Antanas Bukartas, Jonas Wallin.

**Methodology:** Antanas Bukartas, Jonas Wallin.

**Project administration:** Robert Finck, Christopher Rääf.

**Resources:** Robert Finck, Christopher Rääf.

**Software:** Antanas Bukartas, Jonas Wallin.

**Supervision:** Jonas Wallin, Robert Finck, Christopher Rääf.

**Validation:** Antanas Bukartas, Jonas Wallin, Robert Finck, Christopher Rääf.

**Visualization:** Antanas Bukartas.

**Writing – original draft:** Antanas Bukartas.

**Writing – review & editing:** Antanas Bukartas, Jonas Wallin, Robert Finck, Christopher Rääf.

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
