## [Decision Letter · Decision Letter 0]

3 Nov 2020

PONE-D-20-26548

Bayesian algorithm to estimate position and activity of an orphan source utilizing multiple detectors in a mobile gamma spectrometry system

PLOS ONE

Dear Dr. Bukartas,

Thank you for submitting your manuscript to PLOS ONE. After careful consideration, we feel that it has merit but does not fully meet PLOS ONE’s publication criteria as it currently stands. Therefore, we invite you to submit a revised version of the manuscript that addresses the points raised during the review process.

We look forward to receiving your revised manuscript.

Kind regards,

Mohammadreza Hadizadeh

Academic Editor

PLOS ONE

Journal Requirements:

2.In your Data Availability statement, you have not specified where the minimal data set underlying the results described in your manuscript can be found. PLOS defines a study's minimal data set as the underlying data used to reach the conclusions drawn in the manuscript and any additional data required to replicate the reported study findings in their entirety. All PLOS journals require that the minimal data set be made fully available. For more information about our data policy, please see http://journals.plos.org/plosone/s/data-availability.

3. We note you have included a table to which you do not refer in the text of your manuscript. Please ensure that you refer to Table 5 in your text; if accepted, production will need this reference to link the reader to the Table.

Reviewers' comments:

Reviewer's Responses to Questions

**Comments to the Author**

1. Is the manuscript technically sound, and do the data support the conclusions?

Reviewer #1: Yes

Reviewer #2: Yes

2. Has the statistical analysis been performed appropriately and rigorously? 

Reviewer #1: Yes

Reviewer #2: Yes

3. Have the authors made all data underlying the findings in their manuscript fully available?

Reviewer #1: Yes

Reviewer #2: Yes

4. Is the manuscript presented in an intelligible fashion and written in standard English?

Reviewer #1: Yes

Reviewer #2: Yes

5. Review Comments to the Author

Reviewer #1: In this paper the authors developed a Bayesian algorithm to estimate the position of a lost photon (or more specifically a gamma) source. They have performed an extensive experiment by employing multiple detectors in a vehicle to perform on-site measurements. The paper is really interesting and important to the field. I only have couple of minor comments on this manuscript. Therefore, I recommend minor revision of the manuscript before being considered for publication in PLOS ONE. Please see the attached PDF file for my comments.

Reviewer #2: Review report on Manuscript no. PONE-D-20-26548

Title:

Bayesian algorithm to estimate position and activity of an orphan source utilizing

multiple detectors in a mobile gamma spectrometry system

General:

I recommend accepting the publication of the manuscript.

Reasons:

This is a well-written paper on the public health and the environment. The manuscript addresses an important issue, to avoid harm to the public and the environment, in case the loss of ionizing radiation sources. The authors manage to compare the experimental measurements with the stat of art mathematical expressions in cases where most would resort to numerical integration or Monte Carlo approaches.

6. PLOS authors have the option to publish the peer review history of their article (what does this mean?). If published, this will include your full peer review and any attached files.

Reviewer #1: **Yes: **Mehrdad Shahmohammadi Beni

Reviewer #2: **Yes: **Mahmoud Ibrahim Abbas

---

## [Author Response · Author response to Decision Letter 0]

11 Dec 2020

Hopefully, we have made all due corrections to make the article fit the requirements of PLOS ONE.

2.In your Data Availability statement, you have not specified where the minimal data set underlying the results described in your manuscript can be found.

Data is now available in the article supplements.

3. We note you have included a table to which you do not refer in the text of your manuscript.

The citing has been revised and made sure that all of the tables and figures are mentioned in the text.

We have also noticed, that the resolution of our images is significantly reduced when building the PDF in Editorial Manager of PLOS ONE. We are providing images of way better resolution, but have no idea on how to increase the resolution in the built PDF file. We can provide a fully built PDF file embedding all of the figure images at correct resolutions for convenience of the reviewers if needed.

For detailed answers to the reviewers questions, please refer to the separate file provided.

---

## [Decision Letter · Decision Letter 1]

2 Jan 2021

Bayesian algorithm to estimate position and activity of an orphan gamma source utilizing multiple detectors in a mobile gamma spectrometry system

PONE-D-20-26548R1

Dear Dr. Bukartas,

We’re pleased to inform you that your manuscript has been judged scientifically suitable for publication and will be formally accepted for publication once it meets all outstanding technical requirements.

Kind regards,

Mohammadreza Hadizadeh

Academic Editor

PLOS ONE

Reviewers' comments:

Reviewer's Responses to Questions

**Comments to the Author**

1. If the authors have adequately addressed your comments raised in a previous round of review and you feel that this manuscript is now acceptable for publication, you may indicate that here to bypass the “Comments to the Author” section, enter your conflict of interest statement in the “Confidential to Editor” section, and submit your "Accept" recommendation.

Reviewer #1: All comments have been addressed

Reviewer #2: All comments have been addressed

2. Is the manuscript technically sound, and do the data support the conclusions?

Reviewer #1: Yes

Reviewer #2: Yes

3. Has the statistical analysis been performed appropriately and rigorously? 

Reviewer #1: Yes

Reviewer #2: Yes

4. Have the authors made all data underlying the findings in their manuscript fully available?

Reviewer #1: Yes

Reviewer #2: Yes

5. Is the manuscript presented in an intelligible fashion and written in standard English?

Reviewer #1: Yes

Reviewer #2: Yes

6. Review Comments to the Author

Reviewer #1: The authors addressed all my comments very well. For my comment #24, your replies was "No, everything is correct", in the previous version of the paper, the unit for total counting efficiency was shown in (m^s) which was incorrect (check Fig. 9 in your initial submission), however in the new revised version (which is now Fig. 10) is now corrected to (m^2). I have no other comments and I recommend the paper to be accepted for publication.

Reviewer #2: this is a well written manuscript. all required questions have been answered and that all responses meet formatting specifications.

7. PLOS authors have the option to publish the peer review history of their article (what does this mean?). If published, this will include your full peer review and any attached files.

Reviewer #1: No

Reviewer #2: **Yes: **Mahmoud Ibrahim Abbas

---

## [Editor Report · Acceptance letter]

11 Jan 2021

PONE-D-20-26548R1 

Bayesian algorithm to estimate position and activity of an orphan gamma source utilizing multiple detectors in a mobile gamma spectrometry system 

Dear Dr. Bukartas:

I'm pleased to inform you that your manuscript has been deemed suitable for publication in PLOS ONE. Congratulations! Your manuscript is now with our production department. 

Kind regards, 

on behalf of

Dr. Mohammadreza Hadizadeh 

Academic Editor

PLOS ONE